# Magnesiothermic Reduction of Silica: A Machine Learning Study

**DOI:** 10.3390/ma16114098

**Published:** 2023-05-31

**Authors:** Kai Tang, Azam Rasouli, Jafar Safarian, Xiang Ma, Gabriella Tranell

**Affiliations:** 1SINTEF AS, Industry Institute, N-7465 Trondheim, Norway; xiang.ma@sintef.no; 2Department of Materials Science and Engineering, Norwegian University of Science and Technology, N-7034 Trondheim, Norway; azam.rasouli@ntnu.no (A.R.); jafar.safarian@ntnu.no (J.S.); gabriella.tranell@ntnu.no (G.T.)

**Keywords:** magnesiothermic reduction, silica, kinetic barrier, Gaussian process machine, machine learning

## Abstract

Fundamental studies have been carried out experimentally and theoretically on the magnesiothermic reduction of silica with different Mg/SiO_2_ molar ratios (1–4) in the temperature range of 1073 to 1373 K with different reaction times (10–240 min). Due to the kinetic barriers occurring in metallothermic reductions, the equilibrium relations calculated by the well-known thermochemical software FactSage (version 8.2) and its databanks are not adequate to describe the experimental observations. The unreacted silica core encapsulated by the reduction products can be found in some parts of laboratory samples. However, other parts of samples show that the metallothermic reduction disappears almost completely. Some quartz particles are broken into fine pieces and form many tiny cracks. Magnesium reactants are able to infiltrate the core of silica particles via tiny fracture pathways, thereby enabling the reaction to occur almost completely. The traditional unreacted core model is thus inadequate to represent such complicated reaction schemes. In the present work, an attempt is made to apply a machine learning approach using hybrid datasets in order to describe complex magnesiothermic reductions. In addition to the experimental laboratory data, equilibrium relations calculated by the thermochemical database are also introduced as boundary conditions for the magnesiothermic reductions, assuming a sufficiently long reaction time. The physics-informed Gaussian process machine (GPM) is then developed and used to describe hybrid data, given its advantages when describing small datasets. A composite kernel for the GPM is specifically developed to mitigate the overfitting problems commonly encountered when using generic kernels. Training the physics-informed Gaussian process machine (GPM) with the hybrid dataset results in a regression score of 0.9665. The trained GPM is thus used to predict the effects of Mg-SiO_2_ mixtures, temperatures, and reaction times on the products of a magnesiothermic reduction, that have not been covered by experiments. Additional experimental validation indicates that the GPM works well for the interpolates of the observations.

## 1. Introduction

Theoretically, silicon is the metal produced by carbothermic reduction at temperatures higher than 1821 °C [1]. Conventional silicon production has at least two drawbacks: high energy consumption and a negative impact on the environment through CO_2_ emissions. It is estimated that the energy required to produce 1 kg of silicon (230–235 MJ) is about 10 times higher than that for iron (20–25 MJ) [2]. The carbon footprint of the primary production of silicon is also about 10 times higher than that of the ironmaking process. Therefore, it is of great importance to develop a new silicon production process with a friendly environmental footprint.

Silicon can be produced by metallothermic reduction at relatively low temperatures with no CO_2_ emissions. Among the candidate elements for the metallothermic reduction of silica, magnesium possesses several advantages: it is abundant in natural resources, highly reactive to silica, relatively low cost, and is readily manufactured by the carbon-free electrolytic process. Recently, the synthesis of advanced structures of nonmetals by metallothermic reduction has just begun to show its versatile capability, and pioneering examples were reported for the preparation of nanostructured silicon by Bao et al. in 2007 [3]. Since 2007, a large number of investigations have been published in the literature [4,5]. However, fundamental knowledge about the magnesiothermic reduction of silica at relatively high temperatures (above the magnesium melting temperature of 800–1100 °C) is still poorly understood.

Metallothermic reductions have been known for more than two hundred years. They have been widely employed for the industrial production of metals and alloys. It is worth noting that the fundamental mechanisms of metallothermic reduction processes still remain elusive in terms of the energy and mass transfer on the evolving interphases, although metallothermic reductions in nature are simple chemical displacement reactions. In particular, metallothermic reduction reactions take place in multicomponent systems, where phase fusion and separation take place at high temperatures with less controllable reaction rates.

In our previous studies, the magnesiothermic reduction of nature quartz at 1073–1373 K was extensively studied experimentally [6,7]. Reaction schemes with the formation of different layers around an unreacted quartz core were proposed based on experimental observations. It has been found that Mg diffusion through the MgO-based product layer controls the whole rate of magnesiothermic reactions. The amount of heat released from the magnesiothermic reduction of silica is too small to increase the temperature of the entire system significantly. The effects of the Mg to SiO_2_ molar ratio, reaction time, and size of quartz have also been discussed in detail. For example, it was found that the quartz particle size significantly influences the reaction rate. For Mg/SiO_2_ molar ratios of 1 and 2, the primary products are MgO and Si. The reaction rate declines, which is attributed to the limited diffusion rate of Mg through the product layer [6]. When the molar ratios of Mg/SiO_2_ are increased to 3 and 4, the reaction rate significantly accelerates, facilitated by the formation of a large amount of liquid and the cracking of quartz particles [6,7]. The final product of these reactions is a mixture of MgO–Mg_2_Si–Si and MgO–Mg_2_Si–Mg for molar ratios of 3 and 4, respectively. Nevertheless, a general mathematical description of reduction products as a function of the starting materials, temperature, and reaction time has not yet been established.

In this work, thermochemical descriptions of the Mg–SiO_2_ pseudo-binary system are first presented using the commercial software package FactSage [8]. The equilibrium relations determined by the thermochemical software can thus be used as boundary conditions for mathematical descriptions of the magnesiothermic reduction of silica. We then propose a physics-informed Gaussian process machine for the machine learning of the small experimental dataset. Additional experiments are also carried out in order to verify the Gaussian process machine predictions. We aim to provide a mathematical description of the magnesiothermic reduction of silica at relatively high temperatures, where liquid metals can be produced.

## 2. Equilibrium Relations

Thermochemical calculations can help us to figure out the fundamental reactions that take place in Mg–SiO_2_ pseudo-binary systems in composition and temperature ranges of interest. Equilibrium relations also provide the framework for the data mining and machine learning algorithms that are applied in this work.

First, the Mg–SiO_2_ phase equilibrium diagram calculated using the commercial thermochemical databases FactPS, FTOxid, and FSStel in the FactSage software package [8] is shown in Figure 1. Because of the low evaporation temperature of Mg, the gas phase become stable at a relatively low temperature in the Mg-rich domain. The calculated Mg-SiO_2_ phase diagram can be approximately divided into two parts: the left part is for the metal-rich system and the right part is the oxide-rich system. For the magnesiothermic reduction of silica to produce Si metals, we focus on the left part of the calculated phase diagram with respect to Mg/SiO_2_ molar ratios of 2 to 4.

The Si and Mg contents in the liquid metal phase obtained from the magnesiothermic reduction of silica are studied. As shown in Figure 2, the Mg and Si contents in the equilibrium liquid metal phase are almost parallel to the y-axis, meaning that the Mg and Si contents in the liquid alloy phase can be reasonably estimated by the Mg/SiO_2_ molar ratios. It is thus reasonable to assume that a target alloy with a high Si content requires a low Mg/SiO_2_ ratio.

On the other hand, the metal fraction is also an important parameter of magnesiothermic reduction products. Little information exists in the literature on how the reaction conditions affect the formation of different phases quantitatively. Figure 3 shows the liquid metal phase fraction and the total metallic species in the reduction products as functions of the Mg/SiO_2_ ratio at different temperatures. At temperatures below 1100 °C, there are optimal magnesium to silicon dioxide ratios. A higher Mg/SiO_2_ dioxide ratio results in the formation of more liquid metals with a lower silicon content.

A comparison of Figure 2 and Figure 3 shows that producing a high-silicon alloy requires the use of lower magnesium to silicon dioxide ratios, along with relatively high reduction temperatures. As Figure 1 illustrates, when the magnesium to silicon dioxide molar ratio is less than 2, the liquid metal phase becomes unstable. Rasouli et al. [6] conducted experiments with a Mg/SiO_2_ molar ratio of 1, but these results did not factor into our analysis.

## 3. Experimental Results

To verify the calculations and predictions of the Gaussian process machine, two experiments were carried out with an initial Mg/SiO_2_ molar ratio of 3, in addition to the experimental results reported by Rasouli et al. [6,7]. The same raw materials were used in the additional experiments. Reduction experiments were carried out in a gas-tight stainless-steel reactor, as schematically shown in Figure 4. The experimental procedures are described elsewhere [6]. The product phases after reduction were identified by X-ray diffraction and an electron probe microanalyzer (EPMA). The MgO, SiO_2_, Mg_2_SiO_4_ and MgSiO_3_ oxides, and Si, Mg, and Mg_2_Si metallic phases were determined by X-ray diffraction analysis.

Figure 5 and Figure 6 show the EPMA mapping of the sample obtained for magnesiothermic reduction at 1100 °C for 1 h. In Figure 5, the unreduced silica core is obvious. The reduced products around the silica particle consisting of different phases, such as MgO, Si, Mg_2_Si, Mg_2_SiO_4_, and MgSiO_3_, around quartz particles limits Mg diffusion to the central part of the silica particles. For more detail about the product layers around the unreacted silica core, one may refer the descriptions given by Rasouli et al. [6]. On the other hand, Figure 6 shows that a significant amount of liquid alloy forms, causing the silica particle to crack and resulting in a significantly higher reaction rate. The silica particle can be reduced almost completely.

The conventional unreacted shrinking core model is inadequate to mathematically describe the complex reaction schemes involved in the magnesiothermic reduction of silica. Rather than developing a new unreacted core model, we attempt to use artificial intelligence-based machine learning methods to describe the experimental observations. As machine learning algorithms possess a self-learning nature, we anticipate that the prediction accuracy will continually improve as experimental observations at different experimental conditions are input.

## 4. Gaussian Process Machine for the Machine Learning of the Magnesiothermic Reduction of Silica

Machine learning approaches are fast and scalable but ideally require large datasets from which to learn. Obtaining such chemical datasets is a significant challenge, as many are proprietary and not readily available for academic use [9]. Experimental determination of the reaction schemes in the Mg–SiO_2_ system is, on the other hand, a time-consuming and expensive task. High-temperature experiments present a challenge not only in terms of devising a suitable design to prevent side reactions but also in terms of accurately measuring the parameters. In order to apply machine learning to such cases, we need to develop algorithms that are capable of learning from small datasets and be able to interpolate the observations. Here, we apply the Gaussian process machine to this work.

The Gaussian process (GP) is a supervised machine learning method that is used to solve regression and probabilistic classification problems [10,11]. It has recently been applied in computational materials science and chemistry, especially for the regression of atomistic properties and the construction of interatomic potentials, known as the Gaussian Approximation Potential framework [12]. On the other hand, attempts have been made to apply a machine learning approach using “hybrid” datasets to describe complex magnesiothermic reductions. In addition to the small experimental laboratory dataset, the equilibrium relations calculated by thermochemical software and its database have also been introduced as boundary conditions for magnesiothermic reductions, assuming a sufficiently long reaction time. Here, we refer to such training datasets as “hybrid”. The table below provides an example of the hybrid dataset employed in this work (Table 1).

The GP is a nonparametric Bayesian approach that can be used for inference purposes. Rather than inferring a distribution over the parameters of a function, the Gaussian process infers a distribution over the function of interest itself. A Gaussian process defines a prior function that is transformed into a posterior function after some values from the prior distribution have been observed. The Gaussian process machine is desirable for the present case, because it performs quite well in small data regimes, provides highly interpretable results, and automatically estimates the predictive uncertainty.

There are several existing GP libraries: Scikit-learn [13], GPy [14], GPflow [15], and GPyTorch [16]. However, none of them incorporate both the non-negativity and mass conservation constraints into the GP algorithm. We, thus, developed the physics-informed code for the above purposes.

For the implementation of the non-negativity constraint, we can simply apply the square root function to the input array so that the Gaussian process machine can force the results to be non-negative when it inverses the square of the results. To satisfy mass conservation constraints, several approaches can be considered: the non-negative GPR [17,18], Bayesian inference [11,19], and constrained optimization [20]. The easiest way is to introduce a mass balance equation into the kernel function of the GPM. We used a set of linear equations that relate the mole inventories of the species to the mole inventories of the elements in the system to enforce the mass balance in this system. Specifically,
(1)$$\sum_{i}^{m}\sum_{j}^{n}a_{ij}X_{i}=b_{j}$$
where *a_ij_* are the stoichiometric coefficients for the reactants, *X_i_* is the mole fraction of the species, and *b_j_* is the mole inventory of element *j* in the system. This equation ensures that the total number of moles of each element in the system remains constant over time. We incorporated the above linear equations into the composited kernel function. By formulating the mass conservation constraints in this way, the Gaussian process machine will automatically satisfy elemental conservation laws when making predictions.

Overfitting is another crucial concern with the Gaussian process machine. Unfortunately, simple kernel functions, such as the radial basis function (*RBF*), Matérn, and periodic function, are inadequate for avoiding overfitting in the present case. To address this issue, we conducted hyperparameter tuning by considering a variety of composite kernel functions. The results of this tuning process indicate that the following composite kernel can effectively address the overfitting problem:(2)$$f_{kernel}=c\left[RBF+White\right]+\alpha \left(\sum_{i}^{m}\sum_{j}^{n}a_{ij}X_{i}-b_{j}\right),$$
where *RGF* refers to the radial basis function, *White* is the white kernel, *c* denotes the constant kernel with a log-uniform prior from 0.001 to 1000 on that value, and *a* is the scaling factor for the mass balance term. Here, we used the Scikit-learn library [13] to train and predict the GPM. While the above hyperparameters were fine-tuned within a limited range of values, it is crucial to keep in mind that their optimization may not extend to the entire range of composition and temperature variables. A regression score of 0.9665 was obtained using the composite kernel function.

Figure 7a shows the GPM-calculated species for the magnesiothermic reduction of silica at 1173 K for 120 min. Figure 7b compares the GPM calculations with experimental data at 1173 K for 240 min. The GPM results include uncertainties with a 50% standard deviation for the predictive distribution. It is obvious that the GPM yields results that align with the experimental values within the limits of its predicted uncertainty.

An important parameter in metallothermic reduction is the effect of the reduction time on the resulting products. Figure 8 displays the variations in the species calculated by the GPM. Almost all experimental points lie within the limits of the uncertainty of the GPM predictions. It is evident that the predictions given by the GPM show some nonsmoothed variation. The uncertainty given by the GPM also increases moderately in the nonsmoothed parts of the curves. Notably, the GPM predicts the variations in species for Mg/SiO_2_ molar ratios of higher than 4, since there are no experimental results available in this range. The GPM predictions require further experimental validation in the future.

The predictive capability of the GPM was validated through experimental results at an Mg/SiO_2_ ratio of 2.9. Figure 9 shows the results. The experimental results (Mg/SiO_2_ = 2.9) are basically consistent with the fractions of Mg_2_Si, MgO, and SiO_2_ predicted by the GPM. Nevertheless, a significant discrepancy exists between the calculated phase fractions of Mg and SiO_2_ and the experimental results. This may stem from the examination methods used for the samples. While the training phase distributions were obtained from a Rietveld analysis of the reacted samples, the validated phase distributions were obtained from EPMA results. Further experiments are necessary to validate the GPM’s calculations.

Figure 10 presents the GPM’s iso-contour predictions for the phase distributions of Mg_2_Si, Si, Mg, and MgO species, with corresponding experimental data overlaid on the diagrams. The GPM’s predictions for the products of metallothermic reductions are reasonably accurate. However, the abnormal variations in the iso-contour curves of Mg_2_Si and Mg species between 120 and 240 min suggest that further experimental validation is necessary to improve the GPM’s results. This also demonstrates the potential for machine learning to assist in the improvement of experiment designs.

## 5. Conclusions

To develop an alternative method for producing silicon and silicon-based alloys, we conducted experimental and theoretical studies on the magnesiothermic reduction of silica under various conditions, including variation of the Mg/SiO_2_ molar ratio from 1 to 4 and reaction times of 10 to 240 min at relatively high temperatures of 1073 K to 1373 K. Based on the Mg–SiO_2_ pseudo-binary equilibrium relations, the boundary conditions of the metallothermic reduction of silica were defined.

Although the unreacted silica core encapsulated by the reduction products has been found in some parts of laboratory samples, other parts of samples have demonstrated near-complete metallothermic reduction. Laboratory investigation results on the magnesiothermic reduction of silica suggest that the traditional unreacted core model is inadequate to represent such complicated reaction schemes.

An attempt was thus made to apply the machine learning approach to describe the complex magnesiothermic reductions using a hybrid dataset. In addition to experimental laboratory data, equilibrium relations calculated by the thermochemical database were also introduced as boundary conditions for magnesiothermic reduction, assuming a sufficiently long reaction time (in this work, longer than 600 min). Here, we refer to such a training dataset as “hybrid”.

The Gaussian process machine was chosen to be trained by the hybrid dataset due to its advantages when describing small datasets. A composite kernel for the GPM was developed to mitigate the overfitting problem that occurs when using generic kernels. The physics-informed non-negative and mass balance constraints were implemented in the GPM. The GPM almost perfectly reproduced the hybrid dataset with a regression score of 0.9665. The trained GPM was thus used to predict the effects of the Mg/SiO_2_ ratio, temperature, and reaction time on the products of the magnesiothermic reduction of silica, which no experiment has covered. Additional experimental validation indicates that the GPM works well for the interpolates of the experimental observations.

## Figures and Tables

**Figure 1 materials-16-04098-f001:**
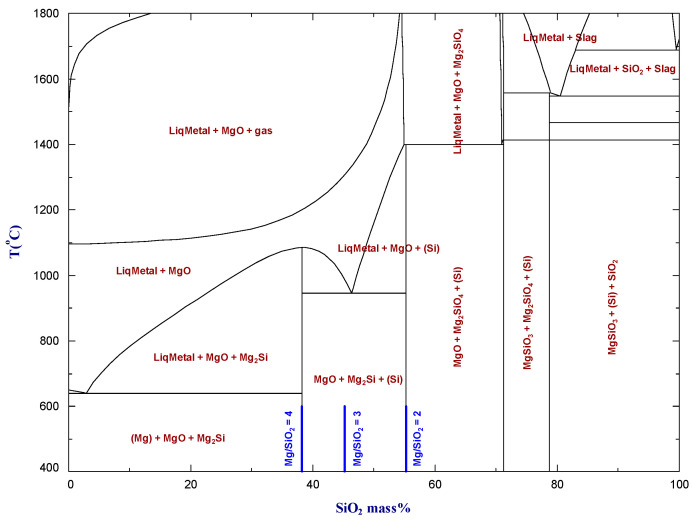
The Mg–SiO_2_ pseudo-binary phase diagram calculated by FactSage and its commercial databases [8].

**Figure 2 materials-16-04098-f002:**
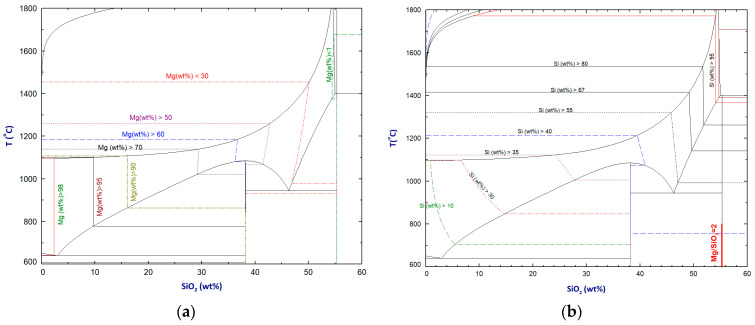
Calculated (**a**) iso-Mg and (**b**) iso-Si content contours in the liquid metal phase.

**Figure 3 materials-16-04098-f003:**
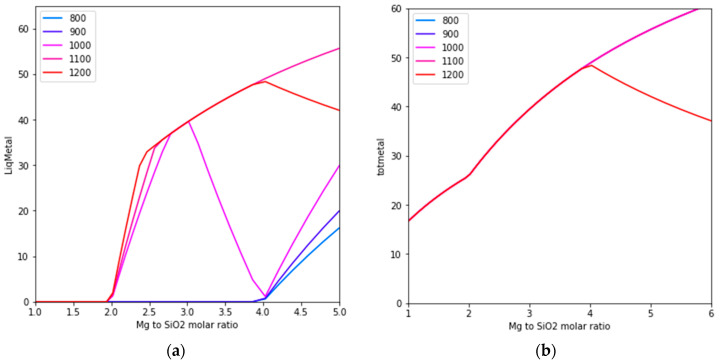
Effect of the Mg/SiO_2_ ratio on (**a**) the equilibrium liquid metal phase fraction and (**b**) total metallic species.

**Figure 4 materials-16-04098-f004:**
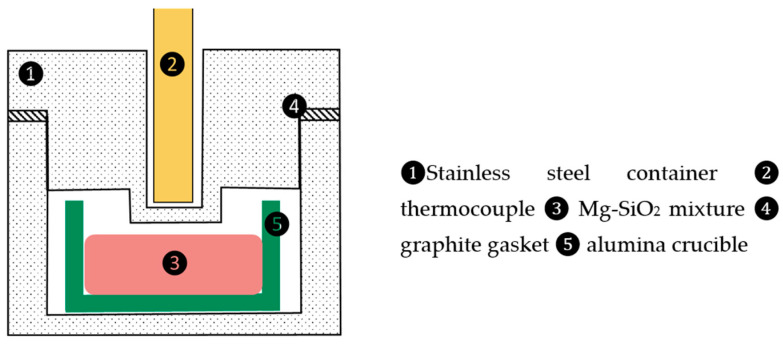
Schematic drawing of the reactor.

**Figure 5 materials-16-04098-f005:**
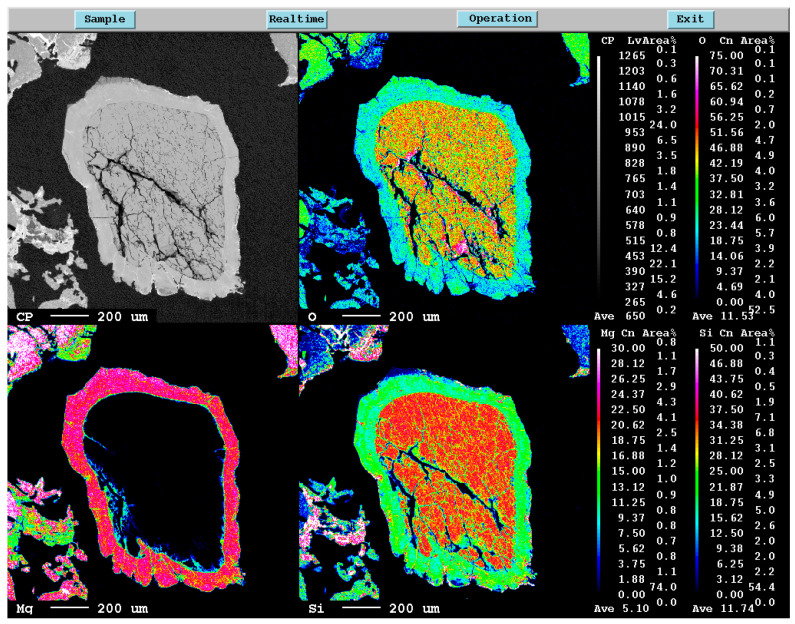
EPMA mapping of Si, Mg, and O elements in the unreacted core sample.

**Figure 6 materials-16-04098-f006:**
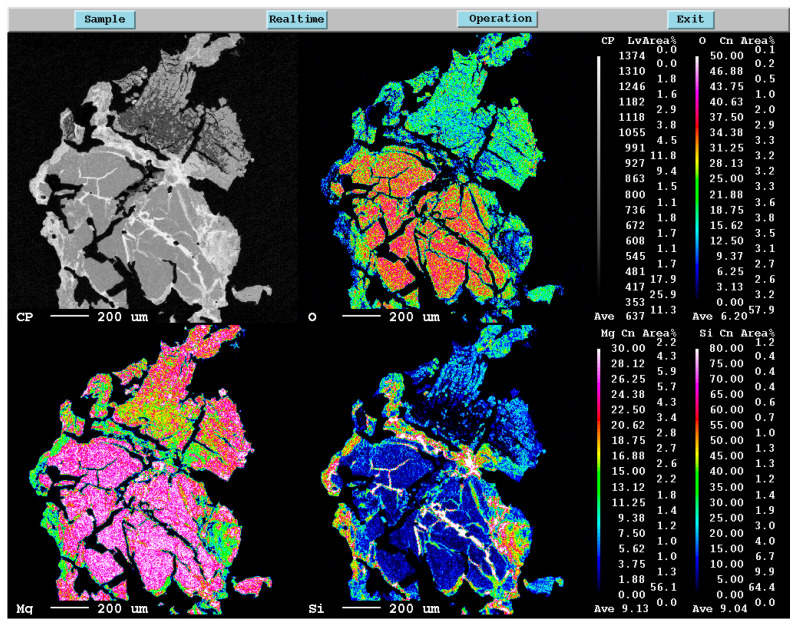
EPMA mapping of Si, Mg, and O elements in the reacted core sample.

**Figure 7 materials-16-04098-f007:**
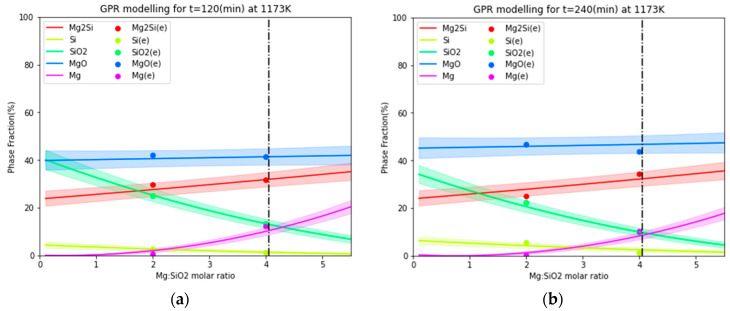
Comparison of GPM-calculated species variation as a function of the Mg/SiO_2_ molar ratio at 1173 K (**a**) for 120 min and (**b**) for 240 min with the experimental results [6,7].

**Figure 8 materials-16-04098-f008:**
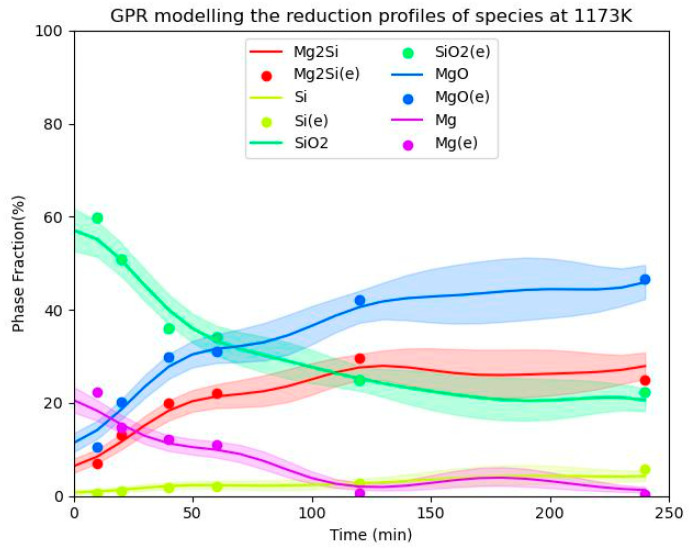
Comparison of the experimental results [6,7] with GPM-calculated species variations over time for an Mg/SiO_2_ molar ratio of 2 at 1173 K.

**Figure 9 materials-16-04098-f009:**
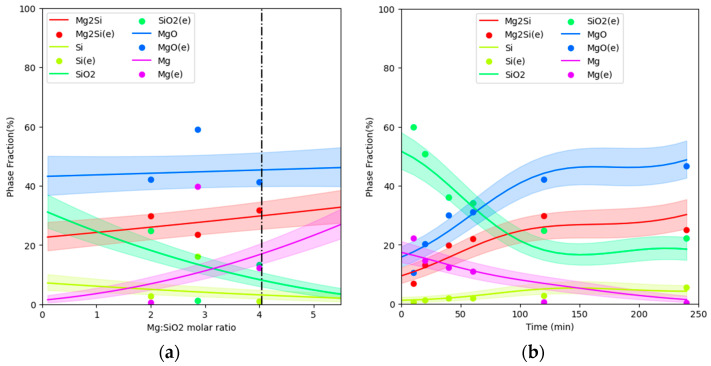
(**a**) Predictive capability of the GPM: phase fractions of species at 1173 K for varying Mg/SiO_2_ molar ratios. (**b**) Comparison of GPM-calculated and measured [6,7] species variations as functions of the reaction time at Mg/SiO_2_ ratios of 2 and 2.9 at 1173 K.

**Figure 10 materials-16-04098-f010:**
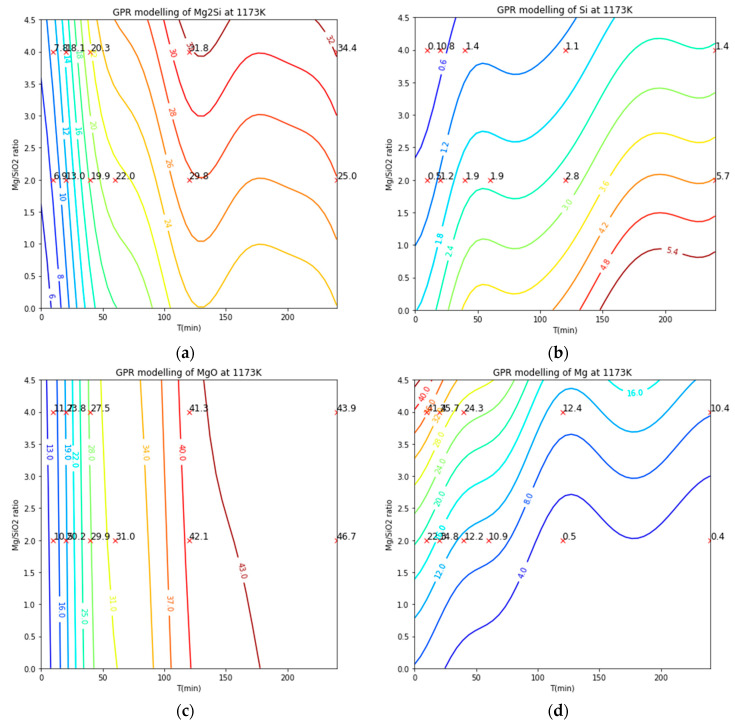
The iso-contours of (**a**) Mg_2_Si, (**b**) Si, (**c**) MgO, and (**d**) Mg generated by the GPM in comparison with the experimental data points [6,7].

**Table 1 materials-16-04098-t001:** Hybrid dataset used for the magnesiothermic reduction of silica *.

Mg/SiO_2_	T (K)	T (min)	Mg_2_Si	Si	SiO_2_	MgO	Mg	Note
2	1173	10	6.87	0.53	59.83	10.5	22.28	Exp.
2	1173	20	13.04	1.18	50.82	20.19	14.76	Exp.
2	1173	40	19.92	1.85	36.07	29.92	12.24	Exp.
2	1173	60	22.02	1.92	34.12	31.01	10.94	Exp.
2	1173	120	29.75	2.76	24.86	42.09	0.54	Exp.
2	1173	240	24.97	5.67	22.25	46.74	0.36	Exp.
2	1373	10	16.05	11.4	16.05	65.4	0	Exp.
2	1373	20	14.9	13.8	14.9	66.6	0	Exp.
2	1373	40	14.7	14.0	14.7	66.9	0	Exp.
	…	…						
4	1173	10	7.8	0.13	38.95	11.72	41.4	Exp.
4	1173	40	20.29	1.45	26.32	27.49	24.32	Exp.
4	1173	120	31.75	1.12	13.43	41.34	12.36	Exp.
4	1173	240	34.37	1.4	9.89	43.9	10.43	Exp.
	…	…						
2	1073	0	0	0	55.28	0	44.72	Des.
2	1173	0	0	0	55.28	0	44.72	Des.
2	1273	0	0	0	55.28	0	44.72	Des.
3	1073	0	0	0	45.18	0	54.82	Des.
3	1173	0	0	0	45.18	0	54.82	Des.
3	1273	0	0	0	45.18	0	54.82	Des.
4	1073	0	0	0	38.2	0	61.8	Des.
4	1173	0	0	0	38.2	0	61.8	Des.
4	1273	0	0	0	38.2	0	61.8	Des.
	…	…						
2	1073	600	0	33.28	0	95.52	0	Equ.
2	1173	600	0	33.28	0	95.52	0	Equ.
2	1273	600	0	33.28	0	95.52	0	Equ.
3	1073	600	45.44	16.64	0	95.52	0	Equ.
3	1173	600	45.44	16.64	0	95.52	0	Equ.
3	1273	600	45.44	16.64	0	95.52	0	Equ.
4	1073	600	90.88	0	0	95.52	0	Equ.
4	1173	600	90.88	0	0	95.52	0	Equ.
4	1273	600	90.88	0	0	95.52	0	Equ.
		…	…					

* The experimental data used in the present work were taken from Rasouli et al. [6,7].

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
