# Peer review of "Magnesiothermic Reduction of Silica: A Machine Learning Study"

_materials, 2023, doi:10.3390/ma16114098_

Round 1

Reviewer 1 Report

2388143 Magnesiothermic review Reduction of Silica: A Machine Learning Study

Review Comments 

1 Graph should be more precise.

2 Data not available for temperature 1373K but mentioned in conclusion.

3 At Page 9, lines 242 and 243, results were examined at a ratio of 2.9, but in Figure 9, it is wrongly mentioned.

4 More references from old research papers only. Kindly add recent work done on the topic.

5 Please provide a reference for the table and graphs used.

1 Graph should be more clear.

2 Data not available for temperature 1373K but mentioned in conclusion.

3 At Page 9, lines 242 and 243 results were examined at a ratio of 2.9 but in Figure 9, it is wrongly mentioned.

4 More references from old research papers only. Kindly add recent work done on the topic.

5 Please provide a reference for the table and graphs used.

Author Response

Thank you for your insightful comments and suggestions. They are indeed valuable in enhancing the quality of our manuscript. Please find below our responses to your feedback:

Graph Precision: We acknowledge your concern regarding the precision of the graph. We are now revising the graph to provide more clarity and improve precision. The axes labels, scale, and data points will be clearly presented.

Data at 1373K: We appreciate you pointing out this discrepancy. The data for 1373K were omitted in Table 1. We have added the omitted data in the Table 1.

Page 9, lines 242 and 243: We apologize for the confusion caused by the mismatch between the text and Figure 9. We have rewritten the captions of Figure 9 with (a) and (b)

Recent References: We appreciate your suggestion to include recent research on the topic. For the experimental part of the relevant work, we use the data that were published in 2022. We cannot find the relevant experimental work in 2023. We are in the process of identifying and including relevant recent publications to provide a more comprehensive and up-to-date context for our study.

References for Tables and Graphs: We realize the need to reference the tables and graphs and apologize for this oversight. We add suitable references to Table 1 and the graphs used in the manuscript.

We are thankful for your constructive comments that help to improve our manuscript.

Reviewer 2 Report

I am attaching the review report of the paper kindly find it.

The quality of the English can be improved.

Author Response

Thank you for taking the time to evaluate our manuscript and provide your invaluable feedback. We appreciate your comments and understand your concerns regarding the clarity of our paper in terms of both the theoretical and experimental aspects of our study. However, we would like to express our view that the study appropriately balances both theoretical and experimental methodologies, as both were vital to our research activities. Our aim was to demonstrate the synergy between theoretical models and experimental procedures in developing a novel method for the metallothermic production of silicon and silicon-based alloys.

Nonetheless, we acknowledge your concerns about the complexity of the material and the potential confusion that it may cause. We certainly aim to ensure our manuscript is clear, succinct, and accessible to all readers. Looking back, we understand how some parts of our work could be confusing. We are grateful for your insight in pointing this out to us.

In response to your comments, we plan to revise the manuscript to improve clarity. This will involve a thorough review of the description of both our experimental procedures and GPM model, with an effort to make our method and results more understandable.

While we respectfully disagree that our manuscript lacks focus, we are committed to taking your feedback into consideration and making appropriate revisions to further improve our work. We hope to strike the right balance that will adequately address your concerns while preserving the integrity of our study.

Once again, we are grateful for your insightful comments, which will ultimately help us improve the quality of our work.

Lines 9-18: We appreciate your feedback and realize that we may need to refine the abstract to make it more concise. Our intention was to provide a brief overview of the study's main themes.

Line 9: By "fundamental studies", we mean to convey that we conducted foundational, essential research into the topic, exploring both experimental and theoretical aspects.

Lines 12-13: The term "well-known thermochemical software" refers to widely recognized and commonly used software in this field of study. We used FactSage, as indicated in the manuscript in the Introduction part.

Line 17-18: This statement is based on our experimental observations. We can include a more detailed explanation or references to substantiate our claim in the revised manuscript.

Line 29: Here we meant to convey that further experimental verification demonstrated the accuracy of the GPM predictions.

Line 36: "Conventional silicon production" is referring to established, widely-used experimental methods of producing silicon.

Line 51-53: We agree that there are many studies on silica production. However, our emphasis was on the magnesiothermic reduction process at high temperatures, which, to our knowledge, has been less explored.

Lines 71-74: Our study includes both theoretical and experimental components, with an emphasis on integrating both aspects for a more comprehensive analysis.

Lines 76-77: The term "additional experiment" refers to supplementary experimental tests conducted to validate our model's predictions.

Lines81-82: This claim is based on established literature and our additional experimental results. We provide detailed explanation and references in the revised manuscript.

Line 110-111 and Line 112-115: We apologize for any lack of clarity in these lines. In the revised manuscript, we have provided a more detailed description and explanation to address this.

Line120-122: The experiments were carried out at different molar ratios to study the behavior under varying conditions.

Lines 60-172: We agree that machine learning ideally requires large datasets, but Gaussian process models can still be effective with smaller, but well-curated datasets. We will better articulate this point in our revision.

Line 240, Lines 245-246, Lines 293 -294: We understand your concern about the credibility of our results. We included this statement to transparently acknowledge that while the GPM predictions were in good agreement with our experimental results, we encourage further experimental validation for more comprehensive verification.

Line 270: We have provided a more detailed description of the experimental and theoretical procedure in the revised manuscript to ensure clarity.

Reviewer 3 Report

1. The abstract should state briefly the purpose of the research, the principal results and major conclusions. An abstract is often presented separately from the article, so it must be able to stand alone. The abstract in the manuscript is complete, but not refined enough.

2. The literature review is acceptable. However, the authors are encouraged to provide an extended review of results reported in other works, discovering the scientific gap regarding the issue and then highlight the necessity and novelty of their own work.

3. In Figure 2(a). Calculated iso-Mg, the author should confirm whether the abscissa is Mg content (wt.%).

4. “The GPM results include uncertainties with a 50% standard deviation of the predictive distribution. It is obvious that the GPM yields results that align with experimental values within the limits of its predicted uncertainties.” The author of Figure 7 in the manuscript needs further explanation and clarification. 50% standard deviation?

5. In Figs7-9, it is suggested that the author further clarify the meanings of dots, lines and colored areas respectively.

6. It is unclear from the manuscript how many times the experiment was repeated, which raises questions about the scientific validity of comparing the experimental data with the results of the GPR calculation.

No

Author Response

Thank you for your time in reviewing our manuscript. We appreciate your valuable comments and suggestions. Here are our responses to each of your comments:

Abstract: We acknowledge your suggestion for refining the abstract. We have revised it to provide a more concise overview of our research purpose, principal results, and major conclusions.

Literature Review: Your suggestion is well-taken. In our revision, we have added more conclusive experimental observations. and more clearly highlighting the novelty and necessity of our study.

Figure 2(a): We apologize for any confusion caused. We have re-plotted the figure to clearly indicate that the abscissa indeed represents Mg content (wt.%).

Clarification of GPM Results: We understand the confusion regarding the statement about the 50% standard deviation. In the revised manuscript, we provide more context and emphasize that it represents the measure of uncertainty in our predictions, not an error in the predictive model itself.

Figs 7-9: We appreciate your suggestion for additional clarity. We have revised the figure captions and possibly incorporate a legend within each figure to clearly identify what the dots, lines, and colored areas represent.

Repeatability of Experiment: We apologize for the lack of clarity on this point. The experimental data used in the manuscript were made by the Ph.D. student, one of the coauthors, Rasouli. She used at least 2 years in her laboratory work.  Conducting high-temperature experiments proved to be rather costly. In contrast, our use of the Gaussian Process Model (GPM) presented a cost-effective solution, as all the libraries employed in this study are freeware. Because of the rapid development of the AIGC, we would expect that the ML modelling will be even cheaper or even free in the near future.

Thank you again for your insightful comments and suggestions. We are confident that these revisions will significantly improve the clarity and quality of our manuscript.

Best Regards,

KT

Reviewer 4 Report

This work is very interesting and well presented and illustrated, properly documented, reliant on appropriate tools and used critically and a real pleasure to read. It rarely happens that I can't propose some at least marginal improvement. As far as I'm concerned this work can be published as it is.

Author Response

Thank you for your kind words and the time you've invested in reviewing our manuscript. We greatly appreciate your positive feedback and are encouraged by your endorsement of the publication of our work.

It is heartening to know that our efforts to present and illustrate our research effectively have been recognized. We also acknowledge the importance of critical tool usage and proper documentation, and we are glad that our work in these aspects has been well received.

We have revised our manuscript according to other reviewers' comments and opinions.

Thank you once again for your support and we look forward to contributing more to the scientific community.

Best Regards,

KT